# Saponins, the Unexplored Secondary Metabolites in Plant Defense: Opportunities in Integrated Pest Management

**DOI:** 10.3390/plants14060861

**Published:** 2025-03-10

**Authors:** Adnan Shakeel, Jewel Jameeta Noor, Uzma Jan, Aabida Gul, Zafar Handoo, Nasheeman Ashraf

**Affiliations:** 1Plant Biotechnology and Molecular Biology Division, CSIR-Indian Institute of Integrative Medicine, Br. Sanat Nagar, Srinagar 190005, Jammu and Kashmir, India; adnanshakeel1717@gmail.com (A.S.); jjameeta.noor744@gmail.com (J.J.N.); januzma586@gmail.com (U.J.); bhat.aabidagul@gmail.com (A.G.); 2Academy of Scientific and Innovative Research (AcSIR), Ghaziabad 201002, Uttar Pradesh, India; 3Mycology and Nematology Genetic Diversity and Biology Laboratory, USDA, ARS, Northeast Area, 10300 Baltimore Avenue, Beltsville, MD 20705, USA; zafar.handoo@usda.gov

**Keywords:** biotic stress, integrated pest management, saponins, secondary metabolites

## Abstract

Plants are exposed to a diverse range of biotic stressors, including fungi, bacteria, nematodes, insects and viruses. To combat these enemies, plants have developed an arsenal of defense mechanisms over time, among which secondary metabolites are the most effective. Moreover, to overcome the negative impact of chemical pesticides, the plant’s secondary metabolites can be harnessed to develop novel disease management strategies. Alkaloids, flavonoids, terpenes and essential oils are major pathogen/pest-responsive secondary metabolite classes in plants. Among these, saponins have shown significant potential in suppressing a wide range of plant pathogens. However, they are yet to be explored thoroughly compared to other secondary metabolites in plant defense, and therefore, a low number of disease control agents exist in agri-markets based on saponins. Thus, this review aims to rectify this bias by identifying and acknowledging the significance of saponins as being on par with other classes of secondary metabolites in plant defense systems. It also provides the first holistic review on the role of saponins with known mechanisms against all of the major plant pathogens/pests. Furthermore, this review discusses the potential of saponin-rich crops in providing eco-friendly pest/pathogen management products for integrated pest management (IPM) and prospectives on the potential of saponin derivatives in developing novel biocides for sustainable agriculture.

## 1. Introduction

With time, plants have evolved in response to pest and pathogen attacks and have developed imperviousness through different tactics. This plant–pathogen interaction, over millions of years, has resulted in the production of more than 200,000 secondary metabolites in plants [1]. These secondary metabolites (SMs) have demonstrated significant importance for plants in mitigating stress conditions and combating biotic pathogens, including insect pests, pathogenic fungi, bacteria, nematodes and viruses [2]. Plant SMs have been classified on the basis of their chemical nature into the major groups of amines, phenolics, terpenoids, alkaloids, quinones, glucosinolates, cyanogenc glucosides, organic acids and polyacetylenes. Saponins, a member of terpenoid SMs, have demonstrated diverse biological activities against various biotic stressors (Figure 1). Different studies have shown the activity of saponins against major herbivorous insects such as aphids, leafworms, termites, weevils and borers [3]. For example, tea saponins have shown the significant suppression of the Diamondback moth (*Plutella xylostella*), a major pest of *Brassica* crops, through its antifeedant and stomach toxicity activities [4]. Saponins have also shown antifungal activities against major plant pathogenic fungi like *Rhizoctonia solani*, *Puccinia recondita*, *Phytophthora infestans*, *Botrytis cinerea*, *Blumeria graminis* and *Colletotrichum coccodes* [5,6,7]. Aescin, a triterpene saponin, exhibits strong antifungal activities against the major *Brassica napus* fungal pathogen, *Leptosphaeria maculans*, by interfering with the sterols of the fungal cell membrane [8]. Saponins also have strong antibacterial properties. Bacoside A, a complex of saponins, has shown significant antibacterial activity against the soft rot pathogen of crops, *Pseudomonas aeruginosa*, by eliminating its biofilm [9]. Moreover, most of the destructive plant parasitic nematodes, such as root-knot, root-lesion and cyst nematodes have been reported to be inhibited by saponin-based formulations [10]. Medicagenic acid saponins have shown considerable nematicidal activity against the plant parasitic nematode *Globodera rostochiensis* by disrupting its cuticle [11]. Although the viricidal activities of saponin have not been well explored in plants, oblonganiside saponins have been reported to have antiviral activities against *Tobacco mosaic virus* by inhibiting the viral replication process [12]. Table 1 provides the latest update on the well-characterized saponins from plants, which signifies the importance of saponins in plant defense.

Despite showing tremendous potential against the major plant pests and pathogens, saponins remain among the least explored plant secondary metabolites in the defense system, and therefore, their application in industry remains unexplored. We confirmed this bias through the bibliometric analysis of articles published during the last decade describing the role of secondary metabolites in the plant defense system through VOSviewer software, version 1.6.20, https://app.vosviewer.com/docs/file-types/map-and-network-file-type/ accessed on 1 May 2024 (Figure 2). Therefore, to rectify this bias, we drafted this review article by using the data from recent research trends, through Web of Science https://www.webofscience.com/wos/woscc/basic-search accessed on 1 May 2024, on the saponin-mediated defense in crops against major insect pests, parasitic nematodes, pathogenic fungi, bacteria and viruses. Only a few reviews exist in the literature [39,40] that demonstrate the significance of saponins against various plant pathogens like fungi and insect pests. However, the other major crop pests/pathogens, like parasitic nematodes and plant viruses, have not been reported in these publications. Moreover, the methodology employed in these reviews is not in accordance with modern academic tools. Therefore, this work is a state-of-the-art review, which brings recent updates with validated methodology on the significance of saponins in the plant defense system. Moreover, this review article is the first holistic approach that discusses the importance of saponins in plant defense against all the major plant pests/pathogens. Importantly, this review identifies the research gaps and suggests the prospects for saponin research aimed towards eco-friendly disease management in crops.

## 2. Methodology

We used scholarly sites like Web of Science and Scopus (https://www.scopus.com/search/form.uri?display=basic accessed on 1 May 2024) to perform the literature survey for this article. A chronological method was adopted by selecting articles published on the role of saponins in plant defense from 2014 to 2024 using the main topic heading of saponins in plant defense and the keywords of plant secondary metabolites, antifungal, insecticide, biocide, nematicide and viricide. The extracted articles were analyzed using VOSviewer software for bibliometric analysis.

## 3. Saponin Biosynthesis in Response to Pest/Pathogen Attack

Saponins are a category of plant specialized metabolites (PSMs), which are produced in plants in response to various biotic stresses. They are characterized by a hydrophobic aglycone backbone attached to hydrophilic saccharide, like glycosides. This dual nature, with both hydrophobic and hydrophilic components, gives these molecules their soap-like and surface-active properties [41]. Based on the aglycone structure, they are classified into three types: steroidal saponins, terpenoid saponins and steroidal glycoalkaloids. There is inadequate data to classify these saponin types specifically against pests/pathogens. However, there is a link between the host plant species, its saponins and the pests/pathogens that attack it. This is evident in *Barbarea vulgaris,* which produces triterpenoid saponins against *Plutella xylostella* [42]. Notably, *Avena sativa* also produces the triterpenoid saponin avenacin against the fungal pathogen *Gaeumannomyces graminis* [20]. These examples make it apparent that the same saponin class is active against different pest/pathogen groups (insect and fungi) in different plant species, and therefore, classifying them against specific groups is difficult. Generally, triterpenoid saponins are found in dicots and steroidal saponins in monocots [43]. Interestingly, all these types have a common precursor, 2,3-oxidosqualene [44,45]. A schematic biosynthetic pathway of saponin biosynthesis in plants is given in Figure 3. The saponin biosynthetic pathway starts with the formation of isopentenyl pyrophosphate (IPP) and dimethylallyl diphosphate (DMAPP) through the cytoplasmic mevalonate pathway (MVA) or the plastid methylerythritol phosphate pathway (MEP) [46]. These precursors then undergo several enzymatic steps to form the basic triterpene or steroidal backbone, which is further modified to form saponins [47]. Through the HMGR pathway, the MVA/3-hydroxy-3-methylglutaryl-CoA reductase converts acetyl-CoA to the five-carbon terpene precursor IPP, which is subsequently isomerized to its allylic isomer dimethylallyl pyrophosphate (DMAPP) by the enzyme isopentenyl diphosphate isomerase [48]. This is followed by the condensation of one DMAPP with two IPP units, producing the 15-carbon farnesyl pyrophosphate (FPP), which is the immediate prenylated precursor of the saponins. Finally, the two FPP units are condensed by squalene synthase (SQS), and this condensation reaction yields the linear 30-carbon precursor squalene. Through epoxidation, squalene is acted upon by the enzyme squalene epoxidase and is converted to 2,3-oxidosqualene. Then, oxidosqualene cyclases (OSCs) induce the cyclization of the 2,3-oxidosqualene to a polycyclic structure, which is the branch point between primary and specialized triterpene metabolism in higher plants [49]. The cycloartenol synthase (CAS) cyclizes 2,3-oxidosqualene to produce the tetracyclic main sterol precursor cycloartenol. Similarly, β-amyrin synthase cyclizes 2,3-oxidosqualene to produce the main triterpene precursor. These precursors then undergo several enzymatic steps to form the basic triterpene or steroidal backbone, which is further modified to form saponins [50]. Generally, the process involves the cyclization of the triterpene/steroid backbone, oxidation, glycosylation and other modifications to produce various saponin compounds. The enzymes mostly included are CYPs and UGTs [51]. CYPs are involved in several key steps of saponin biosynthesis, including the hydroxylation and oxidation of triterpene or steroidal precursors [52]. These reactions are essential for the modification of the precursor molecules into diverse saponin structures. CYPs also catalyze the formation of important intermediates in saponin biosynthesis, such as aglycones and glycosides, through various oxidative reactions. Similarly, UGTs or UDP-glycosyltransferase play a crucial role in saponin biosynthesis. UGTs are responsible for attaching sugar moieties to the aglycone backbone of saponins, which are typically triterpene or steroids. This glycosylation step is essential for the biological activity, water solubility and stability of saponins [53]. Thus, both CYPs and UGTs are involved in the biosynthesis and modification of saponins to produce a wide variety of structures, leading to the diversity of saponins found in plants.

## 4. Role of Saponins in Plant Defense and Integrated Pest Management (IPM)

### 4.1. Saponins in Insect Pest Management

Insects, comprised in the phylum Arthropoda of the animal kingdom, include most of the pests that attack crop plants, causing 20% losses in annual crop yields globally [54]. These arthropod pests pose a threat to all plant stages, including vegetative and reproductive growth, leaves and shoots, through feeding on plant sap or chewing various parts. To deter these insect pests, plants have evolved an astonishing range of secondary metabolites. Saponins, the terpenoid secondary metabolites, have shown promising results in managing these insect pests (Table 2). Along with fungicidal activity, insecticidal potential is the major biological activity of saponins that has been harnessed by researchers over time for integrated pest management [55]. The mechanisms for the insecticidal activity of saponins can be divided into three categories, as illustrated in Figure 4.

#### 4.1.1. Antifeeding or Repellant Mechanism

This mechanism demonstrates that saponin-rich crops or saponin-supplemented artificial diets reduce the food intake in insects, leading to their reduced survival and low reproductive potential. Some studies have also reported that saponins slow down the digestive potential of insects, reducing their survival and ability to engage in herbivory activities. Iovinella et al. [3] demonstrated the antifeedant activity of saponins isolated from alfalfa against the pest *Popillia japonica*. Saponins like zanhic acid, medicagenic acid and soyasaponin 1 were extracted and treated to the leaves of host plants susceptible to *P. japonica*. Further, the food deterrent activity was determined by supplementing the leaves of *Corylus avellana* with isolated saponins. Both of these observations showed the promising antifeedant activity of saponins against the pest *P. japonica*. *Acorus calamus*, an important medicinal plant, has also been reported to have insect deterrent activities against the pest *Sitophilus oryzae*, and GCMS revealed the presence of saponins in its methanolic extracts [73]. Maazoun et al. [74] observed the antifeedant activity of saponins present in *Agave americana* against the rice pest *Sitophilus oryzae*. The triterpene saponins from *Quillaja saponaria* have been reported to have significant deterrent activity against *Acyrthosiphon pisum*, an important pea pest [75]. Four saponins were isolated from *Q. saponaria* belonging to two saponin classes: two triterpene saponins, Aescin and *Q. saponaria* saponins, and two steroidal saponins, diosgenin and digitonin. These saponins were supplemented through an artificial diet to the pea pest *Acyrthosiphon pisum*, and the results showed a strong deterrent activity in the triterpene saponins.

#### 4.1.2. Membrane Permeabilization and Toxicity

At the cellular level, saponins alter the permeability of the plasma membrane in insects, resulting in the lysis of the host cell. Due to their bipolar nature, saponins, with their lipophilic portion, easily integrate into the lipid portion of the plasma membrane along with the other hydrophilic part, which disrupts the normal integrity of the plasma membrane [75]. They further alter the permeability of the mucosal cells in the small intestine, leading to reduced nutrient absorption in insects. Moreover, they reported the cell permeation activity of saponins extracted from *Q. saponaria* on the gossypium leafworm, *Spodoptera littoralis*. The saponins in the tea plant (*Camellia oleifera*) have been reported to kill the larvae of *Plutella xylostella*, a lepidopteran pest of tea plantation, by shortening the intestinal villi and disrupting the intestinal wall of the pest [4]. Some saponins exhibit different modes of action, such as obscurosides, the triterpenoid saponins isolated from *Clematis obscura*. They show antifeeding activity, cuticle toxicity, the permeation of the intestinal villi and damage to cell organelles like the endoplasmic reticulum and mitochondria in two important pests, *A. pisum* and *P. xylostella* [56].

#### 4.1.3. Interrupting Cholesterol Uptake and Molting

Insects are dependent on their food for cholesterol as they cannot synthesize the precursor sterol structures by themselves [75]. Also, these precursors are important for their growth molting hormone 20-hydroxyecdysone. Therefore, insects feed on plants to obtain these precursors for cholesterol and molting hormone biosynthesis. However, the presence of saponins has been reported to block this sterol uptake by insects [76]. These saponins form insoluble complexes with the sterols, preventing their uptake and digestion in insects. Moreover, if insect larvae feed on a saponin-rich diet, the ingested saponin may form complexes and prevent the synthesis of 20-hydroxyecdysone, preventing their molting and development process. The steroidal saponins show structural similarity with 20-hydroxyecdysone and have been reported to exert an antagonistic effect on the ecdysteroid reporter complex, resulting in the disruptive molting and metamorphosis of insects. Sivaramakhrishnan et al. [77] have evaluated this mechanism of saponins against the red flour beetle (*Tribolium castaneum*). This study reported that the saponin extracts of *Crotalaria stipularia* seeds interrupted the cholesterol metabolism that caused the abolishment of egg hatching and resulted in the mortality of the *T. castaneum* pest. In Diamondback moths, tea saponins have been reported to delay molting by interrupting the molting hormone 20-hydroxyecdysone [40].

### 4.2. Saponins for Plant-Parasitic Nematode Management

Plant parasitic nematodes are devastating pathogens, which cause enormous yield losses in crops [78,79]. Globally, annual losses of about USD 157 billion are reported to be incurred in crops as a consequence of parasitic nematode infestation [80]. These pathogens infect different parts of the plant to complete their life cycle. The most predominant classes, root-knot nematodes and cyst nematodes, infect the root system of the host plant, while *Aphlenchoides* establishes itself in the seeds of the wheat host plant. Recently, nematode-infecting leaves have been identified as *Litylenchus* [81]. Similar to plant–microbe interactions, secondary metabolites produced by the host plant are crucial during nematode infection [82]. Saponins, especially triterpenoids, have been found to significantly control *Meloidogyne* spp. on various plant species (Table 3).

The nematicidal activity of saponins has been reported against several classes of major plant parasitic nematodes. Root-knot nematodes, being the most harmful class, have received the most attention from researchers in this context. The saponin-rich genus of the plant kingdom, *Medicago*, has demonstrated nematicidal activity against a wide range of species belonging to the *Meloidogyne* genus, such as *M. heyniana*, *M. arborea*, *M. lupulina*, *M. truncatula*, *M. arabica*, *M. hybrida*, *M. murex* and *M. sativa* [86,94]. Waweru et al. [84] have demonstrated the nematicidal activity in *Embelia chimperi* against *Meloidogyne* spp. due to the presence of saponins. Similarly, several other plant species (*Quillaja saponaria*, *Azadirachta indica*, *Moringa oleifera*, *Lantana camara*, *Glycyrrhiza glabra*, *Pulsatilla koreana* and *Solanum lycopersicon*) have been reported to have considerable nematicidal activities against *Meloidogyne* spp. due to their richness in saponins [92]. After *Meloidogyne*, the second most saponin-targeted nematode group is root-lesion nematodes (*Pratylenchus* spp.). Altinkoy et al. [85] have reported that a nano silver additive aqueous extract containing saponins, prepared from *Moringa oleifera*, inhibited the root-lesion nematode *P. thornei*. Another study reported that a saponin-enriched compost prepared from wild sunflower suppressed the infection of *P. brachyurus* [89]. Triterpene saponins present in *Medicago truncatula* have also been reported to inhibit *P. penetrans* [88]. Apart from these major nematode genera, saponins have also been reported to suppress the infection of other important plant parasitic nematodes, including *Radopholus similis* [83], *Xiphinema index*, *Globodera rostochiensis*, *Tylenchorhynchus* spp., *Criconemoides xenoplax* and *Helicotylenchus* spp. [86,96]. All these studies clearly demonstrate that saponins have great potential to suppress plant parasitic nematodes and could thus be used to formulate eco-friendly biocontrol agents.

#### Mechanism of Nematicidal Activity in Saponins

From the above discussion it becomes evident that saponins exhibit considerable nematicidal activity against a wide range of plant parasitic nematodes. However, the mechanistic insights into the nematicidal activity of saponins have not been comprehensively elucidated yet. Initial studies have revealed that the nematicidal activity of saponins depends on the nature of sapogenin and its side chain sugar molecules [98]. The nematicidal activity in the saponin-rich genus *Medicago* is credited to its sapogenin, medicagenic acid, and two aglycones, hederagenin and bayogenin [99]. At the cellular level, most of the saponins act by altering the cell permeability through interaction with the nematode cell membranes. Some findings also suggest that saponins inhibit the cholesterol synthesis in nematode eggs, leading to their reduced survival [91]. However, the nematicidal activity of saponins varies with nematode species, as reported by D’Addabbo et al. [90]. They found that *G. rostiochiensis* was more sensitive to artemisinin than *M. incognita.* The stage of the nematode life cycle also determined the nematicidal activity of saponins. They also found that the second stage juvenile J2 of *G*. *rostiochiensis* was more susceptible to *Medicago* extracts than *Xiphinema index* and *M. incognita.*

### 4.3. Antifungal Activity of Saponins in Plants

Fungal pathogens impart tremendous pressure on plant produce, causing agriculturists to search for combating strategies to feed the growing global population [100]. Worldwide, growers suffer losses of about 10–25% in crop yields each year, despite the use of fungicides [101]. These mycelial pathogens have developed different strategies to invade plants, and in terms of counter-mechanisms, plants have developed defense mechanisms to combat the effects. This has put huge pressure on both the host as well as the pathogen to devise efficient mechanisms or strategies to win the game. Though plants’ natural defense mechanisms fail to cope with the evolved pathogenic strategies of infection, the manipulation of these mechanisms is the only way to achieve sustainable disease management. Saponin biosynthesis has also evolved in plants in response, and therefore, they have the potential to be used as antifungal agents [102].

Saponins create a chemical barrier between fungi and the host plant, hence protecting plants from fungal diseases [103]. Almost all the major classes of fungi have been found to be inhibited by saponin treatment (Table 4). The tunic layer of saffron corms is a source of toxic antifungal compounds, recently identified as azafrin saponins, against many soil-borne fungi such as *Fusarium exospore*, *Aspergillus niger* and *Bipolaris spicifera*, hence protecting corms against fungal diseases [7]. Like many other crops, the avocado crop (*Persea americana*) is prone to fungal diseases. *Colletotrichum gloeosporioides* is a well-known avocado pathogen. Saponin-rich fractions of the medicinal plant *Enterolobium cyclocarpum*’s bark and *Amphipterygium adstringens*’ branch aqueous extract (ARA) infusion containing the sarsa sapogenin inhibited the growth of *C. gloeosporioides*. However, the strongest inhibition (33%) was observed for *Enterolobium cyclocarpum* bark [104]. *Allochrusa gypsophiloides* is a rich source of triterpenoid saponins and various parts of plants have been evaluated for its effect on various fungal pathogens. Its adventitious root cultures with very high saponin contents have shown growth inhibitory effects against *Saccharomyces cerevisiae* [105]. Quino saponins are recommended as antifungal compounds because of their significant antifungal activity in vivo compared to in vitro assays. Quino saponins at 0.5 to 0.1 g/L solution indicate the induction of resistance in plants. Root soaking with Quino saponins is an effective preventive measure against tomato Fusarium wilt compared to foliar spray and seed soaking [106]. The whole plant of *Allium nigrum* is a source of saponins; however, quantitative variations are found in its different organs. *Allium* root extract, as the richest source of saponins among its other organs, produces Aginoside saponin, a spirostane-type glycoside which shows antifungal activity against *F. oxysporum* and *C. gloeosporoides* in a dose-dependent manner [107]. A recently isolated terpenoid saponin, 3-O-[α-L-rhamnopyranosyl (1–2)-α-larabinopyranosyl]-28-O-[α-L-4-O-acetyl rhamnopyranosyl (1-4)-β-D-glucopyranosyl-(1-6)-β-D glucopyranosyl]-hederagenin), showed very high antifungal activity with MIC values ranging from 0.78 to 100 μg/mL [108]. Methanolic extracts of fruit pericarp and the seeds of *Putranjiva roxburghii* containing saponins have shown antifungal activity against commonly crop-infecting strains of *Aspergillus candidus*, *Chrysosporium tropicum*, *Rhizopus stolonifer*, *Microsporum canis* and *Trychorphyton rubrum*. However, the strongest activity was shown against *Trychorphyton rubrum* [109].

#### Mechanism of Antifungal Activities in Saponins

The formation of complexes between saponin aglycone and the cell membrane sterols of pathogens have been observed, which resulted in pore formation and the loss of membrane integrity. Studies have claimed the inhibition of in vitro mycelial growth in fungi by *Sapindus mukorossi* saponins [110]. Fluorescence labeling and ultrastructural observations have revealed that autophagic-like vacuoles were induced, organellar homeostasis was disturbed and membrane potential and integrity were compromised in *Botrytis cinerea* by saponin treatment. In addition, *S. mukorossi* saponins abolished cell vitality by damaging the normal structures of mitochondria, inducing oxidative burst in the cytoplasm. Oleiferasaponin D3, also known as tea saponin, is a *Camellia oleifera* cake saponin that inhibited the growth of mycelium, decreased cell adhesion and aggregation, and effectively prevented the formation of biofilm. It worked against fungal pathogens such as *S. cerevisiae* and *Penicillium italicum* by breaking down the structure of their cell membrane. This was performed by downregulating the expression of several hyphae- and biofilm-related genes (*ALS3*, *ECE1*, *HWP1*, *EFG1* and *UME6* [111]. Among the many saponins studied, Aescin showed strong antifungal activity. Its antifungal activity was attributed to its ability to interfere with fungal sterols which can be reversed by ergosterol. Aescin triggered plant immune responses in *Brassica napus* and activated plant immunity through the induction of the salicylic acid pathway and oxidative burst, hence protecting plants against *Leptosphaeria maculans* fungi [8]. The crude extract of *Camellia semiserrata* showed antifungal activity against post-harvest pathogens by inactivating fungi like *C. musae*, *C. gloeosporioides* and *C. papaya.* The disease development was controlled by the inhibition of the germination rate, germ tube elongation, distortion rupture and indentation of *C. musae* mycelium (fungus involved in banana anthracnose) and the activation of the defense system of banana trees by the upregulation of defense-related genes *PAL*, *POD*, *GLU* and *CHT* [117]. Steroidal saponins from the genus *Allium* reduced glucose utilization rates and decreased catalase activity and protein content in fungi [114].

### 4.4. Bactericidal Potential of Saponins in Plants

In addition to other pathogenic agents, the bacterial diseases with the highest ranking in spreading infections are another threat to crop production. In contrast to other pathogens, bacterial infection is a rapid process and symptoms like wilting and drooping have been observed in plants within a day. In plants, six genera of bacteria have been reported to cause disease, which include *Agrobacterium*, *Erwinia*, *Corynebacterium*, *Pseudomonas*, *Streptomyces* and *Xanthomonas* [118]. This problem has been successfully addressed by synthetic antibiotics; however, new approaches to treating infections are needed due to the fast-rising rate of resistance to antibiotics. Therefore, screening natural products with antibiotic effects is one of the potential ways to achieve cost-effective and efficient molecules acting against these deadly plant bacterial diseases. Saponins are one such group of secondary metabolites produced by plants, which have shown a significant inhibitory effect on different pathogenic bacteria [119]. Different root extracts enriched with saponins extracted from *Viola pilosa* have shown antibacterial effects [120]. Butanol root extracts of *V. pilosa* have shown antibacterial effects against *Xanthomonas campestris.* Ginsenosides are triterpenoid saponins reported from *Panax ginseng*. To elucidate the role of ginsenosides in *Panax* against its bacterial pathogens, the *PgSTS* gene expression in ginseng organs as well as the monoterpene and sesquiterpene contents from ginseng seedlings treated with elicitors have been analyzed. Salicylic acid (SA) and methyl jasmonate (MeJA) have triggered *PgSTS* expression at different time points and significantly induced the monoterpene and sesquiterpene yields. The results were further confirmed by overexpressing *PgSTS* in *Arabidopsis*, which induced a high terpene content and conferred tolerance against *Pseudomonas syringae* pv. tomato infection [121]. *Silene vulgaris* contains oleanane-type saponins. Hairy root cultures of *S. vulgaris* have been established by infecting leaf explants with five strains of *Agrobacterium rhizogenes* (LBA9402, R1000, A4, 13333 and 15834) to check for the production of saponins [122]. Five- and two-fold accumulation of segetalic acid and gypsogenic acid (triterpenoid saponins), respectively, occurred after MeJA treatment compared to the control root, indicating the role of saponins in defense through the jasmonic acid pathway. Therefore, hairy root cultures of *S. vulgaris* could be a potential source for saponin production, which can further be used as a biopesticide. Among the many saponins tested, Aescin has been effective in the inhibition of bacterial colonies, activating the defense gene phytohormone and activating reactive oxidation species production by impinging on (SA)-dependent immune mechanisms without showing any direct antibacterial effect on the *A. thaliana-Pseudomonas syringae* pv. tomato DC3000 pathosystem. These findings support the use of Aescin as a bactericide [8].

In addition to direct bactericide activities, saponins can alter the bacterial diversity in the rhizosphere soil zone surrounding the roots of plants. Studies have been carried out using four different pure saponin compounds [123]. All saponin treatments decreased bacterial α-diversity and caused significant differences in β-diversity when compared with the control. Saponin treatments depleted certain bacterial taxa (18 families) while others were enriched; however, there were more depleted taxa than enriched ones (Burkholderiaceae and Methylophilaceae). Sphingomonadaceae were enriched in soil treated with α-solanine, dioscin and soyasaponins, which otherwise were abundant in the rhizosphere of saponin-producing plants (tomato and soybean). α-Solanine and dioscin (steroid-type aglycone) specifically enriched Geobacteraceae, Lachnospiraceae and Moraxellaceae, while soyasaponins and glycyrrhizin (oleanane-type aglycone) did not specifically enrich any of the bacterial families. The effects of gypsophila saponins on the growth kinetics of rhizosphere bacteria have been studied. Gypsophila saponins (1%) increased the lag phase of bacterial growth. A pot experiment of subterranean clover rhizosphere revealed no modification of the clover biomass with 1% gypsophila saponin, while there was a two-fold increase in the weight of the adhering soil. The number of culturable heterotrophic bacteria of the clover rhizosphere was not affected by the addition of gypsophila saponins, while qualitative and quantitative differences in the dominant Gram-negative strains were induced by 1% saponins. The addition of saponins changed the dominant populations in the clover soil. *Chryseomonas* spp. significantly decreased, whereas *Aquaspirillum dispar* increased and *Aquaspirillum* spp. became the major genus. The results were supported by knowing that *Aquaspirillum dispar* and *Aquaspirillum* spp. were present as the dominant rhizosphere bacteria of *Gypsophila paniculata* [124]. In vitro analysis of tomatine and tomatidine treatments of field soils have shown the enrichment of twenty and two families, respectively, and the depletion of thirty-five and seventy-eight families, respectively. The bacterial communities of tomato rhizosphere soils were similar to that of the tomatine and tomatidine treatments of field soils, with the complete dominance of the Sphingomonadaceae family and enrichment of the genus *Sphingobium* in particular [125]. These results indicate the specificity of saponins towards different bacterial communities, which can be exploited for the manufacture of biopesticides and the plant production or crop improvement programs.

### 4.5. Saponins for Viral Disease Management in Crops

Viruses form an important group of plant pathogens which cause significant losses in crops. The antiviral role of saponins has been mostly focused on animal pathogenic viruses. However, some studies have reported the viricidal activities of saponins on plant viruses. The saponins extracted from *Panax notoginseng* have shown inhibitory activity against *Cucumber mosaic virus* [126]. *Tobacco mosaic virus*, a major plant virus, has been found to be inhibited by the triterpenoid saponins known as oblonganosides which were isolated from the leaves of *Ilex oblonga* [12]. Saponins have also shown indirect viricide activities by eliminating the virus vectors, as reported by Pensec et al. [127]. They found that saponins extracted from roots of *Gypsophila paniculata* eliminated the nematode vectors (*Xiphinema index* and *X. diversicaudatum*) that were transmitting *Grapevine fanleaf virus* and the *Arabis mosaic virus.* In this way, the elimination of vectors resulted in management of the viruses. In *Chenopodium quinoa*, bitter varieties, due to a high saponin content, have been reported to enhance resistance against the *Cucumber mosaic virus* when compared to the sweet varieties with a low saponin content [126]. The mechanism of antiviral activity in saponins mainly involves the interruption of viral attachment with the host, the inhibition of replication and the regulation of immune response signaling pathways [12].

### 4.6. Saponins for Integrated Pest Management

From the above discussion, it has become quite evident that saponins are effective against a wide range of pests and pathogens in crops. However, there are variations in plants with respect to saponin content. Some plant species are rich in saponins, as illustrated in Figure 5. The high-saponin-containing crops include *Glycyrrhiza glabra* with a 22.2–32.3% saponin content, *Yucca schidigera* with 10%, *Quillaja saponaria* with a 9–10% saponin content, *Beta vulgaris* (5.8%), *Primula* spp. (5–10%), *Saponaria officinalis* (2–5%), *Crocus sativus* (1.2–3.4%) and *Avena sativa* (0.1–0.2%) [128]. These saponin-rich plants offer a great opportunity for developing eco-friendly products in IPM. QL Agri^35^ is a saponin-based eco-friendly nematicide and acaricide available on the market for the IPM of soil-borne pathogens. It is developed from *Q. saponaria* by BASF Mexico. SaponAID^®^ is one more saponin-based product available on the market for IPM. Moreover, saponin-based nano particles are being investigated against plant parasitic nematodes [85]. Recent trends indicate that synthetic or semi-synthetic saponins are being investigated by exploiting their structure–activity relationship to improve efficiency and increase the diversity of saponin-based products [129,130]. However, there is insufficient scientific evidence about their eco-friendly nature, and thus, more efforts are needed for the synthesis of eco-friendly saponin derivatives.

## 5. Conclusions and Future Prospectives

Saponins have demonstrated significant inhibitory activities against most plant pests/pathogens. From herbivorous insects to pathogenic fungi, bacteria, nematodes and plant viruses, saponins have considerably restricted their pathogenicity through various mechanisms. This review highlights the potential of plant-derived saponins in managing major crop pests and pathogens so that more attention can be drawn towards developing saponin-based biocides in the agri-market. Further, this review identifies the research gaps and provides recommendations that can be considered in the future to harness the role of saponins in defending plants against major pests/pathogens in a more sustainable way. These recommendations include the following:The identification of pathogen-specific saponin genes in plants can provide new insights into their role in plant defense.Gene editing tools like CRISPR Cas9 can be used to upregulate saponins in crops to fight against major pathogens.Through breeding techniques, saponin-derived resistance can be introduced to susceptible crop varieties.The integration of saponins in formulations against major pests/pathogens can be performed to develop novel biocides for IPM.More attention should be given to elucidating the mechanistic insights of the saponin-mediated rhizosphere microbiome against soil-borne plant pathogens/pests.More investigation is needed to evaluate the environmental safety of synthetic or semi-synthetic saponins.

## Figures and Tables

**Figure 1 plants-14-00861-f001:**
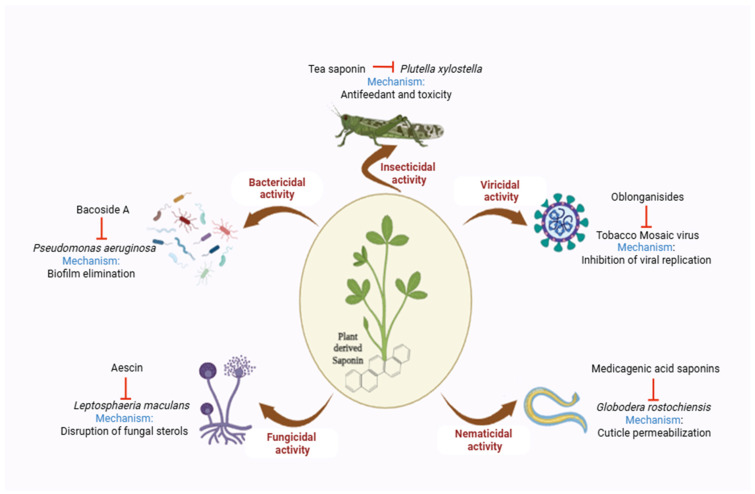
Role of saponins in plant defense.

**Figure 2 plants-14-00861-f002:**
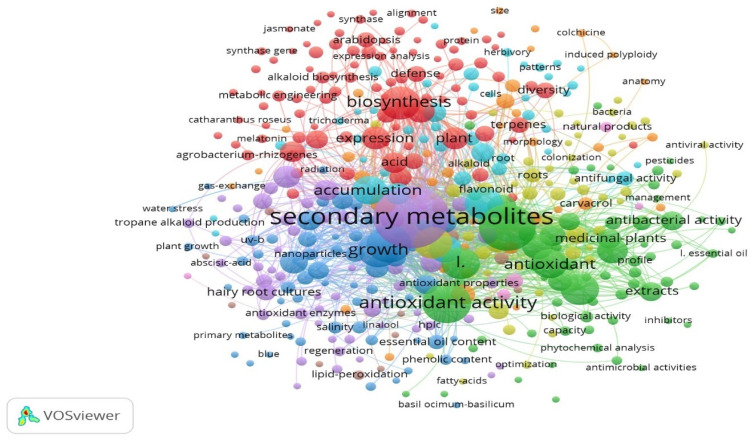
Bibliometric analysis of the role of secondary metabolites in plant defenses during the last decade based on selected keywords such as secondary metabolites, plant defense, biocides, bioinsecticides, biofungicides and pest management.

**Figure 3 plants-14-00861-f003:**
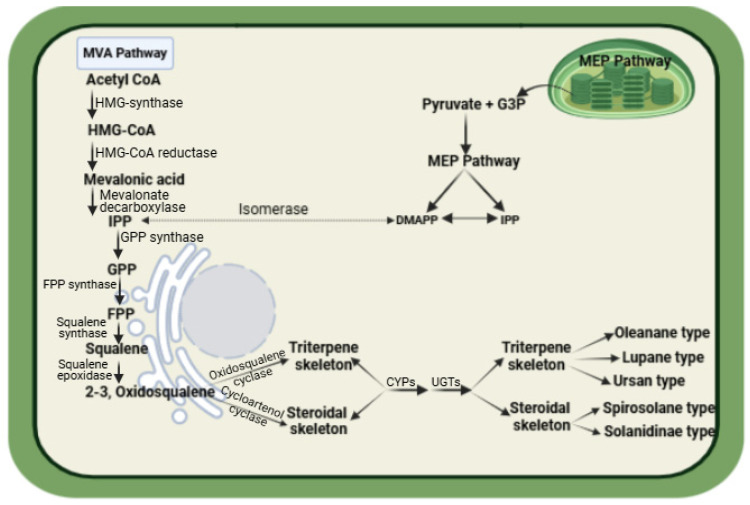
Schematic representation of saponin biosynthetic pathway in plants.

**Figure 4 plants-14-00861-f004:**
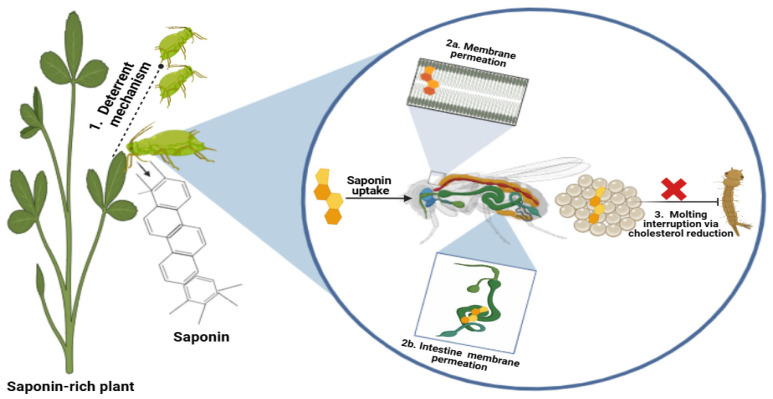
Insecticidal mechanism of saponins in crops.

**Figure 5 plants-14-00861-f005:**
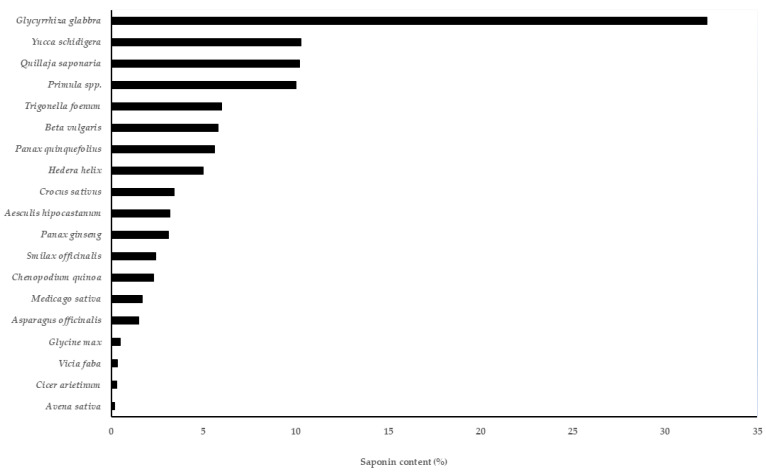
Saponin content in different plant species.

**Table 1 plants-14-00861-t001:** An up-to-date list of saponins isolated and characterized from plant species.

Saponin	Class	Source	Family	Reference
Soyasaponin-1	Triterpene saponin	*Glycine max*	Fabaceae	[13]
Azafrine 1 and 2	Triterpene saponin	*Crocus sativus*	Iridaceae	[14]
Centelloside, Asiatic acid, Madecassoside, Asiaticoside	Triterpene saponin	*Centella asiatica*	Apiaceae	[15]
Escinla	Triterpene saponin	*Aesculus* *hippocastanum*	Sapindaceae	[16]
Gypsogenin, Norhederagenin	Triterpene saponin	*Beta vulgaris*	Amaranthaceae	[17]
Hederagenin	Triterpene saponin	*Hedera helix*	Araliaceae	[18]
Soyasaponins-A, B, E	Triterpene saponin	*Cicer arietinum*	Fabaceae	[19]
Avenacin	Triterpene and steroidal saponin	*Avena sativa*	Poaceae	[20]
Clematichinenosides AR (2)	Triterpene saponin	*Clematis chinensis*	Ranunculaceae	[21]
Mussaendoside	Triterpene saponin	*Mussaenda pubescens*	Rubiaceae	[22]
Saikosaponin	Triterpene saponin	*Bupleurum chinense*	Apiaceae	[23]
Ursolic acid, Oleanolic acid	Triterpene saponin	*Melissa officinalis*	Lamiaceae	[24]
Gymnemasaponin, Gymnemic acid, Gurmarin	Triterpene saponin	*Gymnema sylvestre*	Apocyanaceae	[25]
Medicagenic acid, Zanhic acid, Soyasapogenol, Hederagenin 9	Triterpene saponin	*Medicago sativa*	Lamiaceae	[26]
Saporin	Triterpene saponin	*Saponaria officinalis*	Caryophyllaceae	[27]
α-Tomatine	Steroidal saponin	*Lycopersicon esculentum*	Solanaceae	[28]
Diosgenin	Steroidal saponin	*Dioscorea villosa*	Dioscoreaceae	[29]
25(S)-5β-Spirostane-1β,3β-diol and 2,3-secoporrigenin	Steroidal saponin	*Trigonellafoenum graceum*	Fabaceae	[30]
Solanine, Chaconin	Steroidal saponin	*Allium nigrum*	Lilliaceae	[31]
Capsicoside G	Steroidal saponin	*Solanum tuberosum*	Solanaceae	[32]
Sarsasapogenin, Shatavarin IV	Steroidal saponin	*Asparagus racemosus*	Asparagaceae	[33]
Dioscin	Steroidal saponin	*Allium paradoxum*	Amaryllidaceae	[34]
Sarsasapogenin, Smilagenin	Steroidal saponin	*Yucca schidigera*	Asparagaceae	[35]
Digitalin	Steroidal saponin	*Digitalis purpura*	Plantaginaceae	[36]
Smilscobinosides A and B	Steroidal saponin	*Smilax scobinicaulis*	Smilaceae	[37]
Spirostanol, Furostanol	Steroidal saponin	*Tribulus terrestris*	Zygophyllaceae	[38]

**Table 2 plants-14-00861-t002:** Insecticidal saponins in plants and their mode of action.

Saponin	Source	Target Pest	Mode of Action	Reference
Obscurosides	*Clematis obscura*	*Acyrthosiphon pisum*	Antifeedant, membrane permeation, toxicity	[56]
Medicagenic acid Zanhic acid	*Medicago sativa*	*Popillia japonica*	Antifeedant	[3]
Terpenoid saponins	*Clematis aethusifolia*	*Plutella xylostella*	Antifeedant and stomach toxicity	[4]
Ginsenosides	*Panax ginseng*	*Plutella xylosteila*	Antifeedant	[57]
Aescin, *Q. saponaria* saponins, diosgenin, digitonin	*Quillaja saponaria*	*Acyrthosiphon pisum*	Antifeedant	[58]
*Q. saponaria* saponin	*Quillaja saponaria*	*Drosophila melanogaster*	Antifeedant	[59]
Triterpene saponins	*Chenopodium quinoa*	*Pseudaletia impuncta*	Antifeedant	[60]
Steroidal saponins	*Drimia pancration*	*Stegobium paniceum*	Antifeedant	[61]
Tea saponins	*Camellia oleifera*	*Ectropis obliqua*	Intestinal membrane permeation	[62]
Crotalaria seed saponins	*Crotalaria stipularia*	*Tribolium castaneum*	Molting inhibition	[63]
Spiroconazole A	*Dracaena arborea*	*Aedes albopictus*	Molting inhibition	[64]
Zygophylloside S (1)	*Zygophyllum coccineum*	*Aedes aegypti*, *Culex quinquefasciatus*	Toxicity	[65]
Clematograveolenoside A (1)	*Clematis graveolens*	*Coptotermis homii*	Toxicity	[66]
Crude saponins	*Atriplex laciniata*	*Heterotermes indicola*, *Monomoriurn pharaonic*, *Tribolium castaneum*, *Rhyzopertha dominica*	Toxicity	[67]
Crude saponins	*Periploca hydaspidis*	*Ribolium castaneum*, *Rhizopertha dominica*	Toxicity	[68]
Crude saponins	*Isodon rugosus*	*Tribolium castaneum*, *Rhyzopertha dominica*, *Monomorium*, *pharaonic*, *Heterotermis indicola*	Toxicity	[69]
Crude saponins	*Polygonum hydropiper*	*Tribolium castaneum*, *Rhyzopertha dominica*	Toxicity	[70]
Steroidal saponins	*Trillium govanianum*	*Aphis craccivora*	Toxicity	[71]
Crude saponins	*Periploca hydaspidis*	*Ribolium castaneum*, *Rhizopertha dominica*	Toxicity	[72]

**Table 3 plants-14-00861-t003:** Nematicidal activity of saponins in plants.

S. No.	Saponin Source	Target Nematode	Reference
1	*Chromolaena odorata*	*Radopholus similis*	[83]
2	*Embelia schimperi*	*Meloidogyne incognita* *Pratylenchus zeae*	[84]
3	*Moringa oleifera*	*Pratylenchus thornei*	[85]
4	*Medicago heyniana*, *M.hybrida*, *M. lupulina*, *M. murex* and *M.truncatula*	*Meloidogyne incognita*, *Xiphinema index*, *Globodera rostochiensis*	[86]
5	*Azaddirachta indica*, *Moringa oleifera*, *Lantana camara* and *Glycyrrhiza glabra*	*Meloidogyne* spp.	[87]
6	*Medicago truncatula*	*Pratylenchus penetrans*	[88]
7	*Tithonia diversifolia*	*Pratylenchus brachyurus*	[89]
8	*Artemisia annua*	*Globodera rostiochiensis*, *M. incognita*	[90]
9	*Solanum lycopersicon*	*Meloidogyne* spp.	[91]
10	*Pulsatilla koreana*	*Meloidogyne incognita*	[92]
11	*Quillaja saponaria*	*Meloidogyne* spp.	[93]
12	*Medicago sativa*	*Meloidogyne* spp.	[94]
13	*Medicago arborea*	*Xiphinema* spp.	[95]
14	*Quillaja saponaria*	*Xiphinema americanum*, *Meloidogyne hapla*, *M. ethiopica* *Pratylenchus thornei*, *Tylenchorhynchus* spp., *Criconemoides xenoplax*, *Helicotylenchus* spp.	[96]
15	*Chlorella vulgaris* (*Algae*)	*Meloidogyne incognita*	[97]

**Table 4 plants-14-00861-t004:** Antifungal activity of saponins in plants.

Saponin	Source	Target Fungi	Mode of Action	Reference
*Sapindus mukorossi* saponin	*Sapindus mukorossi*	*Botrytis cinerea*	Impairing membrane integrity and organellar homeostasis	[110]
*Allochrusa gypsophiloides*	*Allochrusa gypsophiloides*	*Saccharomyces cerevisiae*	Immunostimulant	[105]
Oleiferasaponin D3	*Camellia oleifera*	*Saccharomyces cerevisiae*, *Penicillium italicum*	Obliteration of cell membrane structure, leakage of cell contents, inhibition of the growth of the mycelium and reduced cell adhesion and aggregation	[111]
Cucurbitacins B	Cucurbits	*Fusarium oxysporum*	Immunomodulator	[112]
Hongguanggenin, Tigogenin, Agavegenin, Hecogenin, Chlorogenin	*Cestrum nocturum*	*Aspergillus fumigatus*	Disruption of cell membrane and inhibition of hyphal formation	[113]
(1→2)-α-l-arabinopyranosyl hederagenin and acyclic sesquiterpene oligoglycoside	*Sapindus saponaria*	*Colletotrichum musae*, *C. gloeosprioides*, *C. boninense*	Inhibition of enzyme fumarate reductase	[108]
Aginoside saponin	*Allium nigrum*	*Fusarium verticillioides*, *Botryotinia squamosa*	Disruption of cell membrane	[107]
Persicosides A and B	*Allium ampeloprasum*	*Penicillium italicum*, *Asperigillus niger*, *Trichoderma harzianum*	Inhibition of glucose metabolism and disruption of cell membrane	[114]
Aqueous extract	*Asparagus officinalis*	*Asperigillus niger*	Inhibition of fungal growth	[115]
Aescin	*Aesculus hippocastanum*	*Leptosphaeria maculans*, *Pseudomonas syringae*	Activation of immune response provides SA-dependent resistance	[8]
Avenacins A 1	*Avena sativa*	*Gaeumannomyces graminis* var. *tritici*	Disruption of cell membrane	[20]
Flabelliferin B	*Borassus flabellifer,*	*Pseuodomonas aeroginosa*	Enhances defense-related secondary metabolites in host	[104]
Sarsasapogenin	*Enterolobium cyclocarpum * *Amphipterygi* spp.	*Colletotrichumgloeosporioides*	Disruption of cell membrane	[104]
Ingadosides	*Inga sapindoides*	*Plasmopara viticola*	Disruption of cell membrane	[116]

## Data Availability

Data are contained within the article.

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
