# Peer review of "Saponins, the Unexplored Secondary Metabolites in Plant Defense: Opportunities in Integrated Pest Management"

_plants, 2025, doi:10.3390/plants14060861_

Round 1
Reviewer 1 Report
Comments and Suggestions for Authors
Review plants Jan 2025
Saponins, The Unexplored Secondary Metabolites in Plant Defense: Opportunities in Integrated Pest Management (IPM)
The article discusses the multi-faceted aspects of plant saponins in pathogen/pest control, multiple mechanisms and prospects in Integrated Pest Management.
In recent years, plant secondary metabolites represent promising agents in plant disease management and as biocontrol agents.
I have some suggestions for the improvement of the manuscript.
Figure 1. Role of saponins in plant defence. A higher resolution image is required. In addition, bioactivities of plant saponins are mentioned: it is suggested to include the mechanisms of fungicidal, viricidal, insecticidal, etc. should be discussed in terms of type of saponin, active concentration, case study etc. for a better understanding.
With respect to major classes of saponins, is there any particular class which is more active than others against pathogens/pests? Discuss.
Figure 3. saponin biosynthesis pathway: The resolution of the image and font size needs to be improved for clarity. The blank space should be reduced.
For IPM, is there any eco-friendly saponin based developed product in the market? Briefly discuss the research in this direction, citing studies if any.
Minor comments:
Keywords: should be in alphabetical order, please rearrange it
Line 14-15: …secondary metabolites are the most decent? What does it mean?
Table content needs to be more aligned.
Please arrange the references as per MDPI guidelines.
Comments on the Quality of English Language
Moderate English revision is required.
Author Response
We are very thankful to the reviewer for thoroughly reviewing our manuscript and providing valuable suggestions regarding the improvement of the manuscript. We have revised the manuscript in response to the comments raised and have highlighted the changes in yellow.
Comment: Figure 1. Role of saponins in plant defence. A higher resolution image is required. In addition, bioactivities of plant saponins are mentioned: it is suggested to include the mechanisms of fungicidal, viricidal, insecticidal, etc. should be discussed in terms of type of saponin, active concentration, case study etc. for a better understanding.
Response: We have revised the figure 1 as suggested. We have also added bioactivity and mechanism in the figure and revised introduction section accordingly. Please see Figure 1 and lines:43-60.
Comment: With respect to major classes of saponins, is there any particular class which is more active than others against pathogens/pests? Discuss.
Response: We have added this information in the revised manuscript. Please see lines: 118-126
Comment: Figure 3. saponin biosynthesis pathway: The resolution of the image and font size needs to be improved for clarity. The blank space should be reduced.
Response: We have revised the figure as suggested. Please see Figure 3.
Comment: For IPM, is there any eco-friendly saponin based developed product in the market? Briefly discuss the research in this direction, citing studies if any.
Response: We have added the saponin based ecofriendly products that are available in market and discussed the research in this direction. Please see lines: 486-494 and Figure 5.
Minor comments:
Keywords: should be in alphabetical order, please rearrange it.
Response: We have rearranged keywords as suggested.
Line 14-15: …secondary metabolites are the most decent? What does it mean?
Response: We have revised it with “most effective” for better understanding.
Table content needs to be more aligned.
Response: We have edited the tables for better alignment as suggested.
Please arrange the references as per MDPI guidelines.
Response: We have arranged the references ass suggested by using the Mendeley referencing tool for MDPI Plants.
Reviewer 2 Report
Comments and Suggestions for Authors
This paper is a review of the subject matter: Saponins, which are naturally occurring plant biochemicals. The paper is very comprehensive in covering the subject matter, and is well written. It should be read and edited for a few grammatical errors.
Author Response
This paper is a review of the subject matter: Saponins, which are naturally occurring plant biochemicals. The paper is very comprehensive in covering the subject matter and is well written. It should be read and edited for a few grammatical errors.
Response: Thanks for providing your valuable feedback on our manuscript. We have corrected the grammatical errors in the revised manuscript as suggested.
Reviewer 3 Report
Comments and Suggestions for Authors
The scope of this review is to update the state of the art on plant saponines and the potential applications of these secondary metabolites in sustainable Integrated pest management strategies of plant diseases and pests.
Overall this is a merely compilative study and the original contribute of the Authors is very poor or even contradictory. In particular the Authors justify this study in view of the fact that there are few reports pertaining the applications of saponins in managing plant diseases and pests. However they highloght that saponins are produced in low amounts by plants (Lines 468-469) and propose synthetic saponins as a sustainable alternative to synthetic chemicals (Lines 477-480; Lines 485-490). Synthetic saponins are not natural substances and do not constitute an alternative to synthetic chemicals: synthetic chemicals that mimic natural substances, e. g. strobilurins fungicides, already exist.
Most of the examples cited in the text concern the in vitro inhibitory activity of saponins against human or animal pathogens (this is consistent with the Authors' premise that applications of these secondary plant metabolites pertain medicine). However the rationale of using them against plant diseases and pests and the potential application methods are not clearly evidentiated.
There are numerous formal inaccuracies in the text: Latin names of species and genera (sometimes reported with the name of the Author and sometimes without; they must be always in italics etc.). Similarly References have to fulfill international standards (e.g. latin names in italics).
In Table 4 (column: Mode of Action) the term 'Inhibition' is very generic.
I expected a critical review of scientific literature and more specific suggestions and proposals.
For more detailed comments see notes in the text (attached PDF file).

Overall the English style could be improved.
Author Response
We are very thankful to the expert reviewer for thoroughly reviewing the manuscript and providing timely valuable suggestions regarding the improvement of the manuscript. We have revised the manuscript in response to the comments raised and have highlighted changes in yellow.
Comment: The scope of this review is to update the state of the art on plant saponins and the potential applications of these secondary metabolites in sustainable Integrated pest management strategies of plant diseases and pests.
Overall this is a merely compilative study and the original contribute of the Authors is very poor or even contradictory. In particular the Authors justify this study in view of the fact that there are few reports pertaining the applications of saponins in managing plant diseases and pests. However they highloght that saponins are produced in low amounts by plants (Lines 468-469) and propose synthetic saponins as a sustainable alternative to synthetic chemicals (Lines 477-480; Lines 485-490). Synthetic saponins are not natural substances and do not constitute an alternative to synthetic chemicals: synthetic chemicals that mimic natural substances, e. g. strobilurins fungicides, already exist.
Response: Thanks for the valuable suggestions. We have revised these sections regarding saponin content in plants as suggested and discussed the important saponins-rich crops, concentration of saponins in them, products that are available based on them and their potential in IPM. Please see lines: 482-495 and Figure 5.
We agree that synthetic saponins do not constitute an alternative to synthetic chemicals and thus we have revised the manuscript accordingly. We have added some saponin based ecofriendly products in the revised manuscript. Further we have suggested more research is required on the environmental safety of saponin derivatives. Please see lines: 487-495, also lines 503-508, 519, 25-28.
Comment: Most of the examples cited in the text concern the in vitro inhibitory activity of saponins against human or animal pathogens (this is consistent with the Authors' premise that applications of these secondary plant metabolites pertain medicine). However, the rationale of using them against plant diseases and pests and the potential application methods are not clearly evidentiated.
Response: Thanks for highlighting this issue. We find such data in section 4.3; we have now removed all the human or animal pathogen based saponin activity in the revised manuscript to improve the rationale of the study as suggested. Please see section revised section 4.3 and Table 4.
Comment: There are numerous formal inaccuracies in the text: Latin names of species and genera (sometimes reported with the name of the Author and sometimes without; they must be always in italics etc.). Similarly References have to fulfill international standards (e.g. latin names in italics).
Response: We have checked and revised the inaccuracies in text and references as suggested.
Comment: In Table 4 (column: Mode of Action) the term 'Inhibition' is very generic.
Response: We have revised the column with more details as suggested. Please see Table 4.
Comment: I expected a critical review of scientific literature and more specific suggestions and proposals.
Response: We have critically reviewed more scientific literature and added new data with saponin-rich crops (Figure 5), their saponin concentration, and ecofriendly products based on saponins, and research prospects on the environmental safety of saponin derivatives.
We have also added more information in response to the comments raised by other reviewers, like revised figures 1 and 2, more discussion on classification of saponins based on pests and pathogens. Please see lines: 118-126.
Comment: For more detailed comments see notes in the text (attached PDF file).
Response: We have revised the manuscript by addressing all the comments noted in the pdf file.